# Effect of Dietary Ramie Powder at Various Levels on the Growth Performance, Meat Quality, Serum Biochemical Indices and Antioxidative Capacity of Yanling White Geese

**DOI:** 10.3390/ani12162045

**Published:** 2022-08-11

**Authors:** Fengming Chen, Jieyi He, Xin Wang, Tuo Lv, Chunjie Liu, Liping Liao, Zibo Li, Jun Zhou, Bingsheng He, HuaJiao Qiu, Qian Lin

**Affiliations:** 1Academician Workstation, Hunan Key Laboratory of the Research and Development of Novel Pharmaceutical Preparations, Changsha Medical University, Changsha 410219, China; 2Institute of Bast Fiber Crops, Chinese Academy of Agricultural Sciences, Changsha 410125, China; 3College of Animal Sciences, Zhejiang University, Hangzhou 310058, China

**Keywords:** ramie, production performance, serum biochemical indices, antioxidative capacity, Yanling white goose

## Abstract

**Simple Summary:**

The present study assessed and proved the potential of ramie as a new feed ingredient in Yanling white geese, expressed as improved production performance, normal organ function as well as a certain promotion on antioxidative capabilities and meat quality in geese.

**Abstract:**

To investigate the effects of different levels of ramie powder (Boehmeria nivea (L.) Gaudich.) (i.e., 0%, 6%, 12% and 24%) on the production performance, serum biochemical indices, antioxidative capacity and intestinal development of Yanling white geese, a total of 256 geese at 56 days of age were randomly divided into four groups and fed a control diet and the control diet supplemented with 6%, 12% and 24% ramie powder, respectively, for 42 days. The results show that dietary supplementation with 12% ramie powder significantly increased the average final weight (*p* < 0.05) and tended to improve the average daily gain (ADG) and feed/gain ratio (F/G) of the test geese (0.05 < *p* < 0.10). Moreover, the dietary inclusion of 12 and 24% ramie powder improved meat qualities by reducing the L* value (*p* < 0.05) and cooking loss (0.05 < *p* < 0.10) of thigh muscle. Compared with the control group, the ramie powder supplementation at different levels increased the serum activities of glutathione peroxidase and glutathione, promoting the antioxidative capacity of the body (0.05 < *p* < 0.10). This study demonstrates that moderate ramie powder is beneficial to the production performance of Yanling white geese and has the potential to be used as a poultry feed ingredient. In conclusion, 12% was the proper supplementation rate of ramie powder in Yanling white geese feed.

## 1. Introduction

To meet the development needs of the livestock industry, China has been importing large quantities of traditional feed resources, such as soybean meal and corn, every year. Domestically, there are abundant unconventional feed resources. Accordingly, it is necessary and practicable to develop new high-quality feed resources to better improve the quality of animal products. Ramie (Boehmeria nivea L. Gaud), well known as “Chinese grass”, is a fiber-yielding plant of the Urticaceae family [1]. The nutritional value of ramie is comparable to alfalfa, and its young stems and leaves are rich in vitamins and proteins, with a reasonable amino acid composition [2]. Specifically, essential amino acids account for approximately 44.04% of the total amino acids, with the highest contents being glutamic acid and aspartic acid. The lysine content is approximately 1.02%, accounting for nearly 5.50% of the total amino acids. Amino acid composition and balance are the main indicators in nutritional evaluations. Ramie leaves contain 17 amino acids, with a total amino acid content of 18.36% [3]. In addition, previous studies had shown that ramie leaves contain a variety of bioactive components, which have antioxidant, blood lipid-lowering, and intestinal development effects [4,5,6]. Therefore, it is practical to assess the impact of ramie as a new feed ingredient in poultry.

China, the largest geese-breeding country in the world, sold 639 million commercial geese in 2020. Geese are able to make full use of the crude fiber in their diets, because of their unique digestive characteristics. The length of the digestive tract of geese is 10 times the length of their body, which causes food in the digestive tract to remain longer; meanwhile, nutrients in the food being digested are more fully absorbed. According to Han et al., crude fiber is required in a goose’s diet to form a reasonable diet structure [7]. However, to our knowledge, using ramie powder in goose feed has rarely been reported. The objective of this study was to evaluate the effects of different levels of dietary ramie powder on the production performance, meat qualities, serum biochemical indices and antioxidative capacity of Yanling white geese.

## 2. Materials and Methods

All experimental procedures were conducted in accordance with the Chinese guidelines for animal welfare and approved by Changsha Medical University.

### 2.1. Ramie Powder Preparation

The leaves and tender tops were cut and collected when the ramie plants (Boehmeria nivea cv. Qingsizhu No.1) grew to approximately 80 cm and then dried immediately at 60 °C for 4 days in a forced air oven until the water content dropped to 7%. The leaf–stem ratio of the dried ramie was 3.37. Then, the dried stems and leaves were crushed into ramie powder using a grinder equipped with a 1.5 mm sieve. The processed ramie powder was then packed in seized bags and kept in a light-resistant place until further use. The chemical constituents of the ramie powder (analyzed value) were as follows: crude protein: 16.84%; crude fiber: 18.7%; calcium: 3.3%; total phosphorus: 0.3%, lysine: 0.64%, and methionine: 0.04%.

### 2.2. Animals, Diets, and Experimental Design

A total of 256 healthy Yanling white geese at 56 days of age were selected and randomly divided into 4 groups, with 8 replicates of 8 birds in each group. Group I was the control group without ramie powder in their diet (0.00%), while groups II, III and IV were treated with dietary supplementations of 6.00%, 12.00% and 24.00% ramie powder, respectively.

The experiment was conducted at the testing base of the Institute of Bast Fiber Crops, Chinese Academy of Agricultural Sciences in Yuanjiang, Hunan Province. The test area was disinfected one week before the experiment, and the goose house was an isolation area with an automatic manure removal device, and three automatic drinking devices were built in each area. During the test period, the geese were fed and watered freely, the health condition of the geese was observed, and the amount of feed consumed was recorded daily. The design of the test diet formulas referenced the National Research Council’s (NRC) own for geese (1994) in combination with experience in the practical production of Yanling white geese as shown in Table 1.

### 2.3. Growth Performance

During the test, the health condition and growth of the test geese were observed every day, and the feed consumption was recorded. The fasting weight of the test geese was weighed in the morning at 98 days age (fasting 12 h in advance), and the average initial weight, average final weight, ADFI, ADG and feed-to-weight ratio (F/G) of the test geese in each group were calculated.

### 2.4. Organ Indices

At the end of the experiment, one goose from each replicate, close to the mean weight of that replicate, was randomly selected for slaughter after 12 h of fasting. The intestine (contents removed), spleen, heart, liver, gizzard, kidney and pancreas were taken and weighed. The organ indices were calculated as a proportion of the body weight.

### 2.5. Meat Qualities

The previously slaughtered test geese’s breast and thigh muscles were separated and physical indices, such as drip loss (24 h), cooking loss (45 min), meat color (45 min), pH (45 min) and shear force (45 min), were measured separately. Meat colors were determined as L*, a* and b*, which were used to indicate the brightness, redness, and yellowness of the muscle, respectively, using a colorimeter (CR400, Minolta Camera Co., Osaka, Japan). The pH values of the pectoral and thigh muscles at 45 min after slaughter were measured with a pH meter (Model 340, Mettler-Toledo GmbH, Zurich, Switzerland). The drip loss at 24 h postmortem and muscle cooking loss at 45 min after slaughter used a method previously described [8]. The meat pieces after measuring cooking losses (minimum of 6 cm × 3 cm × 3 cm) were sheared at a speed of 1 mm/s with a C-LM3B digital tenderizer (C-LM3B, Northeast Agricultural University, Harbin, China).

### 2.6. Measurement of Intestinal Indices

One goose from each replicate that was close to the average weight of the replicate was randomly selected for slaughter at the end of the test. After slaughtering, the intestine was removed, and the duodenum, jejunum and ileum were immediately separated. Approximately 2 cm of the middle sections of each intestine were taken, gently washed with saline, and then dried with filter paper and fixed in 10% formaldehyde phosphate buffer. After conventional dehydration, paraffin embedding, sectioning and H&E staining, the crypt depth and villi height of the intestines were measured in 10 fields of view from each section, using Motic Images Advanced 3.2 software, to calculate the villi height/crypt depth (V/C) values.

### 2.7. Serum Biochemical Indices and Antioxidant Biomarkers

At the end of the test, one goose from each replicate that was close to the average body weight of that replicate was randomly selected for venous blood collection from the wing. Five milliliters of blood samples were left for 30 min at room temperature, centrifuged at 3000 r/min for 15 min, and then the serum samples were stored at −20 °C and used for the determination of biochemical indices.

The contents of total protein (TP), albumin (ALB), alkaline phosphatase (ALP), alanine aminotransferase (ALT), aspartate aminotransferase (AST), glucose (GLU), triglycerides (TG), total cholesterol (TCHO), urea nitrogen (BUN) and creatinine (Cr) in serum were measured using an automatic biochemical analyzer (URIT-8000, Guilin URIT Medical Electronics Co., Guilin, China). The uric acid (UA) content was measured with a test kit (Zhejiang Dongou Diagnostic Products Co., Wenzhou, China). The ratio of ALB to globulin (GLB) was calculated, while the GLB content was obtained by calculation, i.e., GLB content = TP content—ALB content.

Total antioxidant capacity (T-AOC), total superoxide dismutase (T-SOD), glutathione peroxidase (GSH-Px), catalase (CAT) and malondialdehyde (MDA) in serum were measured using detection kits (Nanjing Jiancheng Bioengineering Institute, Nanjing, China).

### 2.8. Statistical Analysis

After preliminary processing of the test data with Excel 2013 software, one-way analysis of variance (ANOVA) was performed to test the homogeneity of variances via Levene’s test, and the normality was tested via the Kolmogorov–Smirnov test using SPSS 25.0 statistical software. If the differences between groups were significant, multiple comparisons were performed using Duncan’s method, with a level of significance of *p* < 0.05, while 0.05 < *p* < 0.1 was considered a trend. The test results are expressed as the mean ± standard deviation. In addition, orthogonal polynomial contrasts were used to analyze the linear and quadratic effects of the different ramie powder levels in the diets on each index of the Yanling white geese.

## 3. Results

### 3.1. Growth Performance

The effects of the different levels of ramie powder on the production performance of Yanling white geese from 56 to 98 days of age are shown in Table 2. The average body weight of the test geese at 98 days was significantly increased by the supplementation with 12.00% ramie powder (*p* < 0.05) and linearly correlated to the different levels of ramie powder in the diets (*p* = 0.016). Meanwhile, the addition of 12.00% ramie powder to the diets tended to improve the ADG and F/G of the test geese (0.05 < *p* < 0.10). Conversely, there was no significant difference in the ADFI of the test geese between each group (*p* > 0.05).

### 3.2. Proportion of Organ to Body

The effects of the different levels of ramie powder on the organ percentage of 98 days old Yanling white geese are shown in Table 3. Supplementation with ramie powder tended to reduce the liver percentage of the test geese compared to the control (0.05 < *p* < 0.10). In addition, there were no significant differences in the other organ percentages among the groups (*p* > 0.05).

### 3.3. Meat Qualities

The effects of different ramie powder levels on the meat qualities of 98 days old Yanling white geese are shown in Table 4. There were no significant differences (*p* > 0.05) in breast muscle quality among the groups. Meanwhile, compared with the control, the test geese whose diets were supplemented with 12.00% and 24.00% ramie powder showed a significant decrease in thigh muscle brightness (*p* < 0.05) and a tendency to reduce the cooking loss of thigh muscle (0.05 < *p* < 0.10). Moreover, a linear negative correlation was observed between thigh muscle brightness and different ramie powder levels (*p* = 0.005).

### 3.4. Intestinal Mucosal Morphology

The morphological structure of the intestinal mucosa of Yanling white geese is shown in Table 5. There were no significant differences in the villi height, crypt depth and villi height/crypt (V/C) depth value of the duodenum, jejunum and ileum among the test groups (*p* > 0.05).

### 3.5. Serum Biochemical Indices

The effects of diets with different levels of ramie powder on the serum biochemical indices of 98 days old Yanling white geese are shown in Table 6. The test groups with diets supplemented with 6.00%, 12.00% and 24.00% ramie powder tended to increase the serum total cholesterol and albumin levels compared to the control (0.05 < *p* < 0.10). Moreover, there was no significant differences in the serum glucose, triglyceride, uric acid, urea nitrogen, creatinine, total protein, globulin, alkaline phosphatase, alanine aminotransferase and aspartate aminotransferase levels as well as the albumin/globulin and alanine aminotransferase/aspartate aminotransferase values among the groups (*p* > 0.05).

### 3.6. Antioxidative Capacity

The effects of the diets with different levels of ramie powder on the serum antioxidant indices in 98 days old Yanling white geese are shown in Table 7. Compared with the control, there was a tendency towards an increase in the serum glutathione peroxidase and glutathione levels in the test geese with different levels of ramie powder in their diets (0.05 < *p* <0.10). However, there were no significant differences in the serum superoxide dismutase, catalase, total antioxidant capacity and malondialdehyde levels among the groups (*p* > 0.05).

## 4. Discussion

Ramie, an herbaceous perennial that belongs to the Urticaceae family, Boehmeria genus, is one of the most important economic crops in south China, and it has great potential for development as a protein feed ingredient because of its fast growth, high yield, nutritious young stems and leaves, high crude protein content and balanced amino acid composition. Domestic Chinese geese are famous for their tender meat and rich nutritional value [9]. 

Ramie powder used as a supplement in traditional feed material is considered a provider of crude protein and crude fiber in diets. Among the diet composition of all groups, no differences in the nutritional or ingredient levels in the diets were designed, except for the contents of corn, soybean meal, rice husk and ramie powder. Li et al. found that diets supplemented with 9% and 12% ramie reduced the final body weight and ADG in finishing pigs compared to the control diets, but the ADFI did not change [10]. However, Lin et al. pointed out that the addition of 6% and 12% ramie to the diet can significantly increase the final body weight and ADG of Linwu ducks, but the addition of ramie had no effect on the ADFI [11]. Similarly, the present study showed that dietary supplementation of 12% ramie significantly increased the final body weight of Yanling white geese compared to the control group. This may be related to the animal’s breed, age and crude fiber levels in the diet. Geese are a grass-fed and grain-saving poultry, and their consumption capacity for crude fiber is better than that of livestock. It was reported that excessive intake of fiber could reduce nutrient digestibility and increase the satiety of birds, leading to a reduction in weight gain [12]. In the present study, as well as Lin’s, the diets of all the experimental groups possessed similar crude fiber contents, while in Li’s study, the crude fiber levels increased as the inclusion levels of ramie increased in the diets, which may account for the different results. Organ percentage is mainly affected by the animal breed and the dietary nutrition levels, which can reflect the total nutritional status of the organism and organ lesions [13]. It was found that supplementation with different levels of ramie powder in the diets had no significant effect on the organ indices of the test geese, which indicates that ramie supplementation in diets would not harm the growth efficacy of the geese compared to the control diets.

Increasing demand for high-quality meat requires new feed formulations and animal breeding to improve meat quality [14]. Tenderness, flavor, juiciness and color are generally considered to be the main factors associated with meat quality [8]. Meat color is the most direct apparent indicator among these factors, which is related to myoglobin and residual hemoglobin as well as the myoglobin content, oxidation status and the ability of light to reflect on the surface. Cooking loss, an important indicator of muscle quality, reflects the water-holding capacity, which not only affects the color, aroma, taste, nutrient content, juiciness, tenderness and other edible qualities of meat, but it also has important economic value [15]. In a previous study, feeding ramie silage to Boer goats can significantly increase the ADG of meat goats, reduce the content of stearic acid in goat meat and improve the meat quality [16]. In our study, supplementation with ramie powder at a certain percentage also improved the muscle flesh color and water retention of Yanling white geese which, in turn, improved the quality of the meat.

The development of nonconventional forage resources requires familiarity with the possible presence of antinutritional factors in addition to the excellent nutritional properties of the forage resources themselves. In a previous study, the growth of rats was stagnant after the addition of 25% ramie leaves to the diet, while mortality in rats occurred after the addition of more than 40% ramie leaves [17], which indicates the existence of an antinutritional factor in ramie. Therefore, the digestion and absorption abilities of the test geese were key indicators in this study, usually measured by villi height, crypt depth and villi height/crypt depth (V/C) [18]. An increase in villi height increases the contact area between the small intestine and nutrients, which facilitates digestion and absorption; a shallow crypt depth means a decrease in the epithelial cell production rate, an increase in the maturation rate and an improvement in the secretion and absorption functions. The V/C value reflects the functional status of the small intestine, and its increase indicates an improvement in the intestinal mucosal structure, which increases the number of villi cells per unit area, promoting digestion and absorption. The results showed that the addition of 6.00%, 12.00% and 24.00% ramie powder to the diets did not significantly affect the morphological structure of the small intestinal mucosa of 56 to 98 days old Yanling white geese, and some of the indicators showed some improvement from the numerical point of view. Under the conditions of this experiment, the nutrient digestion and absorption of the Yanling white geese were not affected by the supplementation with ramie powder, which proves its feasibility and safety as a feed ingredient for Yanling white geese.

Many physiological and biochemical parameters in the blood are influenced by factors such as growth and developmental stages, dietary nutrition levels and endocrine conditions, reflecting the physiological status of the body’s metabolism. In this experiment, the test geese with different levels of ramie powder in the diet showed a tendency to increase the serum total cholesterol and albumin levels. Cholesterol is an important component of various membrane structures and myelin sheaths in animal cells, a precursor of bile acids, steroid hormones and VD3, as well as an essential factor for lipid transfer system in blood, which has important physiological functions. Maintaining a relatively high level of serum cholesterol should be conducive to a high metabolism and the rapid growth of animals, shortening the raising cycle. On the other hand, elevated serum total protein and albumin levels are a sign of vigorous protein metabolism, indicating enhanced protein anabolism in liver and protein deposition in tissues [19]. This suggests that the addition of certain levels of ramie powder into the diets of 56 to 98 days old Yanling white geese under the present experimental conditions can promote the metabolism and absorption of nutrients, thus improving the production performance.

Oxidative stress is mainly caused by the excessive production of reactive oxygen species (ROS), which plays a role in the biological damage affecting animal growth and production. It also causes lipid peroxidation that deteriorates meat tenderness by inhibiting calpain activity and protein hydrolysis processes [20]. In organisms and muscle tissue, the balance between oxidants and antioxidants is regulated by a defense system composed of enzymatic components, such as SOD, CAT and GPX, nonenzymatic compounds, such as vitamin C and GSH, as well [21]. Previous studies showed that dietary supplements containing natural antioxidant components could promote growth, enhance the antioxidant defense system and improve meat and egg quality in poultry [22,23,24]. The antioxidant defense system induced by Nrf2/ECH works by triggering the antioxidant response elements and promoting the expression of antioxidant enzymes in various tissues [25]. When oxidative damage occurs, MDA and 8-OHdG are formed as the main forms of ROS-induced oxidative lesions; thus, they are considered as common biomarkers of oxidative stress [26,27]. Therefore, dietary ingredients that enhance the activities of the antioxidant defense system has been widely used in animal feed to prevent oxidative stress [28]. Ramie is reported to be rich in active natural products, such as chlorogenic acid and flavonoids, which have strong antioxidant activity, promoting the expression of antioxidant enzymes by activating the endogenous antioxidant response pathways, thus improving the antioxidant capacity of the animal [29]. The current study showed that dietary supplementation with different levels of ramie powder tended to increase the serum GSH-Px and GSH levels in the test geese. GSH-Px is a key peroxidase enzyme that is widely present in the body [30], and it is an important antioxidant and free radical scavenger by binding to free radicals and converting harmful toxic substances into harmless ones [31]. The results above are in line with previous studies, having proved the antioxidant efficiency of ramie.

## 5. Conclusions

This study suggests a dietary supplementation of 12% ramie powder can promote the production performance of Yanling white geese with no adverse effect on organ function, while it led to a certain improvement in meat quality, possibly due to the enhancements of the antioxidative capabilities in geese. This study provides solid data for utilizing ramie as a feed ingredient for geese, contributing to enriching the feed sources and alleviating the current situation of source shortages in China.

## Figures and Tables

**Table 1 animals-12-02045-t001:** Composition and nutrient levels of the basal diets (air-dry basis, %).

Item	Diets			
	Control	6.00%	12.00%	24.00%
Ingredients (%)				
Corn	44.92	46.71	48.73	52.54
Soybean meal	25.46	22.96	20.41	15.42
Rice husk	13.34	10.02	6.67	0.00
Ramie powder	0.00	6.00	12.00	24.00
Oil	8.70	7.23	5.61	2.48
Wheat bran	3.00	3.00	3.00	3.00
Limestone	2.10	1.56	1.04	0.00
CaHPO_4_·2H_2_O	1.07	1.07	1.05	1.00
*L*-Lys	0.05	0.08	0.11	0.17
*DL*-Met	0.06	0.07	0.08	0.09
NaCl	0.30	0.30	0.30	0.30
1% Premix ^1^	1.00	1.00	1.00	1.00
Total	100.00	100.00	100.00	100.00
Nutrient Levels ^2^ (%)				
ME/(Mcal/kg)	3.00	3.00	3.00	3.00
Crude protein	15.00	15.00	15.00	15.00
Crude fiber	8.68	8.68	8.68	8.68
Calcium	1.07	1.07	1.07	1.07
Total phosphorus	0.52	0.52	0.53	0.53
Available phosphorus	0.31	0.31	0.31	0.31
Lysine	0.85	0.85	0.85	0.85
Methionine	0.30	0.30	0.30	0.29
Methionine + cystine	0.55	0.55	0.55	0.55

^1^ The premix provided the following (per kilogram of complete diet) micronutrients: VA, 12, 000 IU; VD3, 2, 500 IU; VE, 20 mg; VK3, 3 mg; VB1, 3 mg; VB2, 8 mg; VB6, 7 mg; VB12, 0.03 mg; D-pantothenic acid, 20 mg; nicotinic acid, 50 mg; biotin, 0.1 mg; folic acid, 1.5 mg; Cu (as copper sulfate), 9 mg; Zn (as zinc sulfate), 110 mg; Fe (as ferrous sulfate), 100 mg; Mn (as manganese sulfate), 100 mg; Se (as sodium selenite), 0.16 mg; I (as Potassium iodide), 0.6 mg. ^2^ Nutrient levels are calculated values.

**Table 2 animals-12-02045-t002:** Effect of different ramie powder levels in the diets on the growth performance of Yanling white geese (56–98 day).

Item	Ramie Powder Level				*p*-Value	Linear and Quadratic Effects of Ramie Powder	
						*p*-Value	
	Control	6.00%	12.00%	24.00%		Linear	Quadratic
Body weight at 56 days (g)	2892.00 ± 51.19	2904.00 ± 57.27	2900.00 ± 61.24	2904.00 ± 50.30	0.984	0.776	0.873
Body weight at 98 days (g)	3691.33 ± 95.24 ^b^	3720.00 ± 107.63 ^b^	3850.67 ± 67.30 ^a^	3803.33 ± 69.12 ^ab^	0.036	0.016	0.341
ADG (g)	19.03 ± 2.64	19.43 ± 1.58	22.64 ± 1.90	21.41 ± 2.70	0.072	0.036	0.435
ADFI (g)	173.88 ± 14.71	178.22 ± 24.84	187.15 ± 25.32	182.89 ± 12.32	0.751	0.386	0.640
F/G	9.20 ± 0.64	9.16 ± 0.86	8.24 ± 0.51	8.59 ± 0.50	0.089	0.049	0.515

^a,b^ Within a row, values with different superscript letters mean there were significant differences (*p* < 0.05). ADG, average daily body weight gain; ADFI, average daily feed intake; F/G, feed/gain ratio.

**Table 3 animals-12-02045-t003:** Effect of the different ramie powder levels in the diets on the proportion of the organs compared to the body weight of the Yanling white geese (56–98 day).

Item	Ramie Powder Level				*p*-Value	Linear and Quadratic Effects of Ramie Powder	
	Control	6.00%	12.00%	24.00%		*p*-Value	
						Linear	Quadratic
Spleen percentage (%)	0.09 ± 0.02	0.11 ± 0.02	0.11 ± 0.04	0.13 ± 0.04	0.353	0.084	0.978
Heart percentage (%)	0.69 ± 0.06	0.70 ± 0.03	0.64 ± 0.04	0.66 ± 0.09	0.504	0.334	0.885
Liver percentage (%)	2.06 ± 0.22	1.91 ± 0.23	1.80 ± 0.26	1.68 ± 0.16	0.077	0.011	0.895
Gizzard percentage (%)	3.64 ± 0.54	3.98 ± 0.79	3.61 ± 0.13	3.49 ± 0.26	0.473	0.438	0.314
Kidney percentage (%)	0.48 ± 0.05	0.47 ± 0.08	0.48 ± 0.06	0.45 ± 0.06	0.821	0.498	0.834
Proventriculus percentage (%)	0.36 ± 0.10	0.33 ± 0.08	0.33 ± 0.02	0.32 ± 0.05	0.775	0.355	0.766
Duodenum percentage (%)	0.63 ± 0.14	0.64 ± 0.13	0.67 ± 0.07	0.62 ± 0.08	0.904	0.922	0.566
Jejunum percentage (%)	1.32 ± 0.23	1.34 ± 0.40	1.08 ± 0.25	1.02 ± 0.08	0.172	0.045	0.716
Ileum percentage (%)	1.44 ± 0.29	1.11 ± 0.22	1.20 ± 0.23	1.38 ± 0.30	0.208	0.893	0.046

**Table 4 animals-12-02045-t004:** Effect of the different ramie powder levels in the diets on the meat quality of the Yanling white geese muscle (98 d).

Item		Ramie Powder Level				*p*-Value	Linear and Quadratic Effects of Ramie Powder	
							*p*-Value	
		Control	6.00%	12.00%	24.00%		Linear	Quadratic
Breast Muscle								
45 min meat color	L*	34.02 ± 4.42	29.88 ± 1.08	29.39 ± 6.63	27.80 ± 3.75	0.190	0.046	0.529
	a*	15.04 ± 3.29	16.19 ± 2.32	12.83 ± 0.97	14.61 ± 2.18	0.192	0.333	0.770
	b*	4.71 ± 1.12	4.90 ± 0.69	4.12 ± 0.59	4.15 ± 0.69	0.328	0.140	0.819
Cooking loss (%)	45 min	42.51 ± 1.30	41.68 ± 2.56	40.51 ± 3.63	38.88 ± 3.19	0.236	0.048	0.752
Drip loss (%)	24 h	11.70 ± 3.02	9.95 ± 1.84	11.29 ± 2.22	9.23 ± 2.68	0.388	0.240	0.893
Shear force (kg·f)	45 min	4.01 ± 0.78	3.85 ± 0.74	3.51 ± 0.42	3.73 ± 0.76	0.708	0.407	0.546
pH value	45 min	6.51 ± 0.26	6.37 ± 0.19	6.22 ± 0.22	6.24 ± 0.19	0.173	0.044	0.414
Thigh Muscle								
45 min meat color	L*	35.52 ± 2.03 ^a^	33.00 ± 4.65 ^ab^	29.58 ± 4.19 ^b^	28.87 ± 2.88 ^b^	0.033	0.005	0.579
	a*	16.01 ± 2.44	13.77 ± 2.96	15.56 ± 2.51	14.03 ± 2.74	0.481	0.448	0.770
	b*	4.70 ± 0.91	4.80 ± 1.23	4.60 ± 0.71	3.79 ± 0.82	0.339	0.138	0.297
Cooking loss (%)	45 min	42.42 ± 1.74	42.52 ± 3.35	40.96 ± 3.18	37.56 ± 4.23	0.092	0.024	0.245
Drip loss (%)	24 h	9.10 ± 2.53	8.24 ± 2.75	7.68 ± 1.75	8.65 ± 2.68	0.824	0.704	0.418
Shear force (kg·f)	45 min	5.63 ± 0.72	4.63 ± 0.67	4.60 ± 0.98	5.12 ± 0.79	0.179	0.340	0.050
pH value	45 min	6.30 ± 0.39	6.24 ± 0.21	6.10 ± 0.22	6.27 ± 0.25	0.666	0.685	0.373

^a,b^ Means in the same row with different superscript letters indicate significant differences (*p* < 0.05). L*, lightness; a*, redness; b*, yellowness.

**Table 5 animals-12-02045-t005:** Effect of the different ramie powder levels in the diets on the intestinal mucosal morphology of 98 day old Yanling white geese (μm).

Item	Ramie Powder Level				*p*-Value	Linear and Quadratic Effects of Ramie Powder	
						*p*-Value	
	Control	6.00%	12.00%	24.00%		Linear	Quadratic
Duodenum							
Villus height	632.09 ± 100.10	636.98 ± 32.92	649.59 ± 47.37	611.16 ± 72.71	0.843	0.718	0.488
Crypt depth	136.75 ± 12.89	128.86 ± 9.29	133.10 ± 4.74	125.64 ± 17.04	0.494	0.239	0.969
V/C	4.61 ± 0.47	4.95 ± 0.22	4.89 ± 0.42	4.87 ± 0.10	0.436	0.311	0.265
Jejunum							
Villus height	585.86 ± 70.14	609.54 ± 59.27	618.07 ± 83.13	586.34 ± 64.07	0.842	0.944	0.387
Crypt depth	111.62 ± 17.39	113.63 ± 11.29	120.93 ± 11.87	107.93 ± 8.78	0.451	0.884	0.206
V/C	5.34 ± 1.01	5.37 ± 0.26	5.11 ± 0.47	5.43 ± 0.41	0.852	0.986	0.604
Ileum							
Villus height	643.29 ± 79.21	637.50 ± 20.20	632.86 ± 32.61	643.06 ± 101.60	0.994	0.969	0.794
Crypt depth	131.03 ± 8.78	129.87 ± 10.45	123.56 ± 7.54	124.37 ± 9.52	0.482	0.169	0.814
V/C	4.90 ± 0.29	4.93 ± 0.28	5.13 ± 0.17	5.15 ± 0.57	0.569	0.192	0.994

**Table 6 animals-12-02045-t006:** Effect of the different ramie powder levels in the diets on the serum biochemical indices of 98 day old Yanling white geese.

Item	Ramie Powder Level				*p*-Value	Linear and Quadratic Effects of Ramie Powder	
	Control	6.00%	12.00%	24.00%		*p*-Value	
						Linear	Quadratic
GLU (mmol/L)	9.24 ± 0.50	9.53 ± 0.56	9.42 ± 0.36	10.03 ± 0.50	0.102	0.034	0.460
TG (mmol/L)	1.07 ± 0.16	1.10 ± 0.09	1.03 ± 0.12	1.10 ± 0.15	0.811	0.976	0.741
TCHO (mmol/L)	7.52 ± 0.61	8.23 ± 0.58	8.02 ± 0.60	8.59 ± 0.58	0.072	0.023	0.796
UA (μmol/L)	479.52 ± 50.58	468.31 ± 56.20	485.34 ± 50.00	420.02 ± 53.65	0.231	0.145	0.268
BUN (mmol/L)	0.73 ± 0.15	0.80 ± 0.11	0.71 ± 0.07	0.68 ± 0.03	0.288	0.223	0.254
Cr (μmol/L)	113.34 ± 29.23	129.40 ± 7.15	122.06 ± 8.18	111.40 ± 19.02	0.403	0.724	0.121
TP (g/L)	37.52 ± 2.35	41.54 ± 4.18	38.63 ± 2.26	38.97 ± 1.90	0.183	0.798	0.162
ALB (g/L)	16.34 ± 0.62	18.03 ± 1.03	17.13 ± 1.31	17.88 ± 1.00	0.068	0.086	0.320
GLB (g/L)	21.17 ± 1.74	23.51 ± 3.41	21.50 ± 1.07	21.09 ± 1.04	0.241	0.589	0.154
ALB GLB	0.77 ± 0.04	0.78 ± 0.09	0.80 ± 0.04	0.85 ± 0.03	0.144	0.038	0.313
ALP (U/L)	17.27 ± 3.10	19.93 ± 4.71	20.30 ± 3.50	20.25 ± 5.43	0.640	0.293	0.491
ALT (U/L)	10.59 ± 1.33	8.76 ± 1.64	8.55 ± 1.46	9.66 ± 1.51	0.160	0.331	0.042
AST (U/L)	2.54 ± 0.13	2.37 ± 0.30	2.13 ± 0.22	2.48 ± 0.36	0.119	0.469	0.046
AST ALT	0.24 ± 0.02	0.28 ± 0.08	0.25 ± 0.02	0.26 ± 0.04	0.606	0.809	0.456

GLU, glucose; TG, triglyceride; TCHO, total cholesterol; UA, uric acid; BUN, urea nitrogen; Cr, creatine; TP, total protein; ALB, albumin; GLB, globulin; ALB/GLB, albumin/globulin value; ALP, alkaline phosphatase; ALT, alanine aminotransferase; AST, aspartate aminotransferase; ALT/AST, alanine aminotransferase/aspartate aminotransferase value.

**Table 7 animals-12-02045-t007:** Effect of the different ramie powder levels in the diets on the serum antioxidant indices of 98 days old Yanling white geese.

Item	Ramie Powder Level				*p*-Value	Linear and Quadratic Effects of Ramie Powder	
	Control	6.00%	12.00%	24.00%		*p*-Value	
						Linear	Quadratic
SOD (U/mL)	855.15 ± 88.99	891.31 ± 59.46	927.39 ± 57.22	872.43 ± 70.73	0.430	0.540	0.166
GSH-Px (U/mL)	619.82 ± 74.97	754.45 ± 76.47	734.10 ± 69.16	716.13 ± 77.88	0.052	0.091	0.036
CAT (U/mL)	3.70 ± 0.77	4.56 ± 0.70	3.82 ± 0.72	3.69 ± 0.70	0.218	0.596	0.149
GSH (μmol/L)	18.17 ± 2.75	24.44 ± 3.51	21.36 ± 4.98	18.74 ± 4.32	0.089	0.866	0.024
T-AOC (mmol/L)	0.91 ± 0.13	0.95 ± 0.13	0.92 ± 0.14	0.87 ± 0.10	0.808	0.599	0.430
MDA (nmol/mL)	4.42 ± 0.40	4.21 ± 0.58	3.88 ± 0.27	4.40 ± 0.46	0.226	0.654	0.085

SOD, superoxide dismutase; GSH-Px, glutathione peroxidase; CAT, catalase; GSH, glutathione; T-AOC, total antioxidant capacity; MDA, malondialdehyde.

## Data Availability

Data is contained within the article. The data used to support the findings of this study are available from the corresponding author upon request.

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
