# Peer review of "Effect of Dietary Ramie Powder at Various Levels on the Growth Performance, Meat Quality, Serum Biochemical Indices and Antioxidative Capacity of Yanling White Geese"

_animals, 2022, doi:10.3390/ani12162045_

Round 1

Reviewer 1 Report

The authors aimed to investigate the effects of different levels of ramie powder (0, 6, 12, 24%) on production performance, serum biochemical indices, antioxidative capacity and intestinal development of Yangling white geese. The result showed that dietary supplementation of 12% ramie powder significantly increased the average final weight and tended to improve the average daily gain (ADG) and feed/gain ratio 20 (F/G) of the test geese. Dietary inclusion of 12 and 24% ramie powder improved meat qualities by reducing the L* value and cooking loss of leg muscle. Compared with the control group, ramie powder supplementation at different levels increased the serum activities of glutathione peroxidase and glutathione, promoting the antioxidative capacity of the body. The study demonstrated that moderate ramie powder was beneficial to the production performance of Yanling white goose and had the potential to be used as poultry feed ingredient. In conclusion, 12% was the proper supplementation rate of ramie powder in Yangling white goose’s feed.

The method section was well written and every step was clear to understand and repeat the experiment. Results and Discussion sections were well designed and written.

This manuscript can be accepted after minor revision.

1. Please give some more details about the Ramie nutrient composition in a table (Line 42).

2. for the statistical analysis please verify the normality and homoscedasticity of the data as a proof for ANOVA is true statistics (Line 184).

3. At line 278 “China, the world’s largest geese breeding country” please give some numerical information in the Introduction section.

4. In Conclusion section a paragraph could be written about the possibilities of transferring Ramie to industry.

Author Response

1. 请在表格中提供更多关于苎麻营养成分的详细信息(第 42 行)。

回应:已在 2.1 部分添加

2. 对于统计分析,请验证数据的正态性和同方差性,以证明 ANOVA 是真实的统计数据(第 184 行)。

回复:使用Excel 2013软件对测试数据进行初步处理后,使用SPSS 25.0统计软件进行单因素方差分析(ANOVA),通过Levene检验检验方差齐性,通过Kolmogorov-Smirnov检验检验正态性。

3. 第278行“中国,世界上最大的鹅养殖国”,请在引言部分给出一些数字信息。

回复:已在介绍部分补充:中国是世界最大的鹅养殖国,2020年销售商品鹅6.39亿只。

4. 在结论部分可以写一段关于将苎麻转移到工业的可能性。

回应:本研究表明,日粮中添加12%的苎麻粉可以促进燕岭白鹅的生产性能,对器官功能没有不良影响,同时对肉质有一定的改善,可能是由于鹅的抗氧化能力增强。本研究为利用苎麻作为鹅饲料原料提供了可靠的资料,有助于丰富饲料来源,缓解我国目前的资源短缺状况。

Reviewer 2 Report

The article presents valuable information with consistent results, however, the composition of the ramie that was used for the formulation still needs to be included in the methodology. How was Rami's ME obtained? What values of proteins, amino acids and minerals of the Ramie used? Without this information, it is difficult to confirm whether the feed composition is really adjusted to maintain the same nutrient levels.

Please enter this information

Author Response

The article presents valuable information with consistent results, however, the composition of the ramie that was used for the formulation still needs to be included in the methodology. How was Rami's ME obtained? What values of proteins, amino acids and minerals of the Ramie used? Without this information, it is difficult to confirm whether the feed composition is really adjusted to maintain the same nutrient levels.

Please enter this information.

Response: The chemical constituents of ramie powder have been added in Part 2.1. Please see the attachment about other revision.

Reviewer 3 Report

Line 14 – it would be better to add the latin name of plant used in the test

Line 74 – is there any number of this approval?

Line 114 – organs are showed as a proportion in body weight?

Line 118 – heart

Line 119 – gizzard

Line 122 – what kind of leg muscles was used? Thigh muscle muscles? Drumstick muscles?

Line 123 – what does it mean: cooking loss (45 min.)? analysis were conducted directly after slaughter? Why not typically after 24 hrs when the carcass is completely cooled? Why so uncommon terms for meat quality evluation were chosen? Normally pH is measured after 15 min., 60 min. and 24 hrs from slaughter to control the post-slaughter glycolysis.

Line 133 – the shape and size of samples should be given as well as the velocity of apparatus knife

Line 153 – if all biochemical parameters were analysed using commercial kits it is not necessary to list all of them

Lines 177 – 183 – what kind of analytical equipment was used?

Line 186 – the quotation shoud be added to statistical software according to its producer reciommendation

Table 2 – it is obvious that average values are presented, it should be rather written „body weight at 56 days (g)”and „body weight at 98 days (g)”. Units should be place with the trait (in column), it will allow to unify all tables and make them easier to read. Also it would be much easier to visualize the production effects by calculating the total average body weight gain.

Lines 214-215 – not indexes were calculated but proportions of particular organs, the same in table 3. Unit (%) should stay next to trait, however, it should be clearly stated (in caption) that it was proportion in body weight (?!)

Tables 4, 6, 7 – slash between trait and unit is redundant

Where is index „1” from legend of table 6?,

Line 273 - Authors Ramie?

Lines 279-283 – it is not discussion

Lines 301, 302 – the quotations are incomplete

Lins 349 – 353 – this sentence should to be rephrased to avoid repetitions

Lines 414 – 415 – this conclusion is too strong due to gallinaceous birds may react by other way, especially highly selected ones. Authors did not investigate it and any references which may prove it are cited. What is the practical impact of the research? Why ramen powder should be used in practice?

Authors should try to limit selfcitations only to those about considering the experimental factor

Author Response

Point 1: Line 14 – it would be better to add the latin name of plant used in the test

Response 1: Boehmeria nivea (L.) Gaudich.

Point 2: Line 74 – is there any number of this approval?

Response 2: have noted in the approval document

Point 3: Line 114 – organs are showed as a proportion in body weight?

Line 118 – heart

Line 119 – gizzard

Response 3: The intestine (contents removed), spleen, heart, liver, gizzard, kidney and pancreas were taken and weighed. The organ indices were calculated as a proportion in body weight.

Point 4: Line 122 – what kind of leg muscles was used? Thigh muscle muscles? Drumstick muscles?

Response 4: Thigh muscle

Point 5: Line 123 – what does it mean: cooking loss (45 min.)? analysis were conducted directly after slaughter? Why not typically after 24 hrs when the carcass is completely cooled? Why so uncommon terms for meat quality evluation were chosen? Normally pH is measured after 15 min., 60 min. and 24 hrs from slaughter to control the post-slaughter glycolysis.

Response 5: Under the premise of conforming to the Chinese standard of livestock and poultry meat quality determination (NY/T1333-2007, NY/T1180-2006), it was moderately adjusted due to the poor experimental conditions at that time.

Point 6: Line 133 – the shape and size of samples should be given as well as the velocity of apparatus knife

Response 6: The meat pieces after measuring cooking losses (minimum of 6 cm × 3 cm × 3 cm) were sheared at a speed of 1mm/s with a C-LM3B digital tenderizer (C-LM3B, Northeast Agricultural University, Harbin, China).

Point 7: Line 153 – if all biochemical parameters were analysed using commercial kits it is not necessary to list all of them

Response 7: have revised

Point 8: Lines 177 – 183 – what kind of analytical equipment was used?

Response 8: had been mentioned in Materials and Methods

Point 9: Line 186 – the quotation shoud be added to statistical software according to its producer reciommendation

Response 9: have added

Point 10: Table 2 – it is obvious that average values are presented, it should be rather written „body weight at 56 days (g)”and „body weight at 98 days (g)”. Units should be place with the trait (in column), it will allow to unify all tables and make them easier to read. Also it would be much easier to visualize the production effects by calculating the total average body weight gain.

Response 10: have revised

Point 11: Lines 214-215 – not indexes were calculated but proportions of particular organs, the same in table 3. Unit (%) should stay next to trait, however, it should be clearly stated (in caption) that it was proportion in body weight (?!)

Response 11: have revised

Point 12: Tables 4, 6, 7 – slash between trait and unit is redundant

Response 12: have revised

Point 13: Where is index „1” from legend of table 6?,

Response 13: have deleted the legend

Point 14: Line 273 - Authors Ramie?

Response 14: have revised

Point 15: Lines 279-283 – it is not discussion

Response 15: have deleted

Point 16: Lines 301, 302 – the quotations are incomplete

Response 16: In the present study and Lin’s, the diets in all experimental groups possessed similar crude fibre contents, while in Li’s study, the crude fibre level was raised as ramie inclusion level increased in diet, which may account for the different result.

Point 17: Lins 349 – 353 – this sentence should to be rephrased to avoid repetitions

Response 17: Under the conditions of this experiment, the digestion and nutrients absorption of Yanling white geese was not affected by supplementation with ramie powder, which proved its feasibility and safety as a feed ingredient for Yangling white geese.

Point 18: Lines 414 – 415 – this conclusion is too strong due to gallinaceous birds may react by other way, especially highly selected ones. Authors did not investigate it and any references which may prove it are cited. What is the practical impact of the research? Why ramen powder should be used in practice?

Response 18: This study suggested a dietary supplementation of 12% ramie powder could promote the production performance of Yanling white goose with no adverse effect on organ function, while had a certain improvement on meat quality, possibly due to enhancements of the antioxidative capabilities in geese. This study provided solid information for utilizing ramie as a feed ingredient for goose, contributing to enriching the feed sources and alleviating the current situation of sources shortage in China.

Point 19: Authors should try to limit selfcitations only to those about considering the experimental factor

Response 19: have reduced
